# Short Communication: The need for open-source hardware, software, and data-sharing specifications in geomorphology

Andrew J. Moodie[1], Eric Barefoot[2], Eric Hutton[3], Charles Nguyen[4], Andrew D. Wickert[4,5], and Jeffrey Marr[4]

[1]Department of Geography, Texas A&M University, College Station, TX, USA
[2]Department of Earth and Planetary Sciences, University of California Riverside, Riverside, CA, USA
[3]INSTAAR, University of Colorado Boulder, Boulder, CO, USA
[4]Saint Anthony Falls Laboratory, University of Minnesota, Minneapolis, MN, USA
[5]Department of Earth & Environmental Sciences, University of Minnesota, Minneapolis, MN, USA

**Correspondence:** Andrew J. Moodie (amoodie@tamu.edu)

**Abstract.** Geomorphologists have more data and computational resources available than ever before. Collaboration between researchers specializing in different modes of inquiry (e.g., numerical, experimental, and field-based) often accelerates impactful scientific insights, but tools to facilitate these collaborations are lacking. In this article, we present four challenges to collaboration in the geomorphology community, and provide a framework that addresses these challenges to enable research utilizing the full extent of data and computational resources available today. We report a component of this framework, a newly developed specification for a shareable data schema called *sandsuet*. The schema is designed to accommodate most kinds of rasterized geomorphology data, and makes it easy to package, publish, and share those data. Finally, we present possibilities for community development of resources to address other challenges to collaboration in geomorphology.

## 1 Introduction

Geomorphology has entered a data-rich scientific era (Paola et al., 2006). It is now easier than ever to affordably build custom sensors and data loggers (Wickert, 2014; Margolis et al., 2020); drones are democratizing aerial lidar and photography (Woodget et al., 2017; Carrivick and Smith; Dronova et al., 2021; Śledź et al., 2021); satellite constellations continuously beam images down from orbit (Nagel et al., 2023; Wu et al., 2023); models analyze these data on powerful cloud computing platforms at previously unimaginable scales (Altenau et al.; Feng et al., 2022). Harnessing this newly available power to drive scientific progress requires collaboration, but geomorphologists lack effective tools to collect, describe, organize, and share community data.

We think it is helpful to identify three main modes of inquiry in geomorphology: (i) *Modern observations* are measurements of ongoing Earth-surface processes or the landforms left by past Earth-surface processes (Fisk, 1944; Ielpi and Lapôtre, 2020; Feng et al., 2022; Nagel et al., 2023; Wu et al., 2023). (ii) *Numerical models* are mathematical formulations of surface processes solved using computers (Watney et al., 1999; Paola, 2000; Overeem et al., 2005). (iii) *Physical experiments* are reduced-scale approximations of surface processes that evolve under laboratory control (Paola et al., 2009; Straub et al.; Esposito et al., 2018).

The most impactful geomorphological research combines different modes of inquiry to bridge scales and arrive at universal insights (Parker et al., 1998b, a; Paola, 2000; Parker et al., 2008a, b; Paola et al., 2009). Research teams generally specialize in one or two of these modes; with collaborations, a project can span all three.

However, current software engineering, data management, and research practices amongst different research teams impede collaboration (Hsu et al., 2015; Grieve et al., 2020). Firstly, to quantify landscape change, researchers create *ad hoc* software tools that lack documentation, do not easily port to new study sites, and go unmaintained after publication (Addor and Melsen; Tucker et al., 2022). When other teams want to use the tool or conduct the same analysis, it is often easier to rewrite the software than it is to use the existing code. Secondly, data products produced by one team seldom fit easily into another team's workflow (Peckham et al.; Peckham, 2014). To collaborate between modes of inquiry, research teams often need to write their own data-wrangling protocols every time they use a new source (Peckham et al.; Peckham, 2014). Finally, while tools for numerical modeling and modern observation have been democratized in recent years (Tucker et al., 2022), tools for physical experiments remain concentrated in a few specialized labs across the world.

These barriers in geomorphology and the aligned fields of sedimentology and (broadly) Earth-surface dynamics echo those in other disciplines. For example, climate model intercomparison projects necessitated standardizing data and software inputs, leading to the development of software packages like CMOR (Climate Model Output Rewriter; Mauzey et al., 2025), and to the founding of open-source community hubs for data sharing like the Earth System Grid Federation. Additionally, the hydrology community organizes around CUAHSI (Tarboton et al., 2008), the ecology and evolutionary biology community has a collaborative ecosystem of training and analysis support (rOpenSci; Boettiger et al., 2015), astronomy leverages a common software infrastructure for research (Astropy Collaboration et al., 2022), and cutting-edge earth systems modeling is a truly community-driven effort (Craig et al., 2012). Each of these projects has been successful because it builds upon existing software and formed a robust community of users (Boettiger et al., 2015). For example, the NetCDF file format for multidimensional arrays (Rew et al., 2006) underpins data and software sharing across Earth-science disciplines because it easily handles coordinates and multiple variables in a single file (e.g., Vo et al., 2024; Eaton et al., 2024; NASA GISS, 2025; International GEOS-Chem User Community, 2025; Iris contributors, 2025).

In this article, we report on several insights gathered from a community workshop on needs for collaborations and data reuse in geomorphology. In response to the most urgent of these needs, we establish the *sandsuet* analysis-ready data specification (v1.0.0), which is a concise set of rules designed to facilitate data sharing that we base on NetCDF. Finally, we describe how the data specification fits into a broader project we are calling the *sandpiper* toolchain to develop technologies to address these problems.

## 2 Community needs in geomorphology

We gathered researchers and engineers working in geomorphology and sedimentology at the Saint Anthony Falls Laboratory, Minneapolis, Minnesota, in late April 2025 for a two-day workshop. We sought to understand how community users generate, analyze, and share raster datasets representing geomorphological change, so that we could gauge interest and requirements for

developing standards. Throughout the workshop, we asked participants to consider datasets with which they are currently (or want to be) working. We considered data-generation workflows, data-analysis workflows, and data-sharing workflows.

From these conversations, we identified four key insights regarding the existing barriers to progress in standardizing data generation, analysis, and sharing in geomorphology. First, we need tools to facilitate the complementary but distinct goals of data archival and data sharing. Second, we must build modular and reusable software that interacts with existing tools. Third, we must expand access to hardware platforms for conducting physical experiments. Fourth, building this system as a community requires a mechanism to give credit to its creators.

## 2.1 Data archival and data sharing serve distinct but complementary purposes.

One key insight that arose from our workshop was that a data standard for sharing analysis-ready products should have different requirements than a data standard for archival storage. Data archival prioritizes the long-term preservation of a comprehensive dataset. Thus, archived data must include extensive descriptions of raw data, instruments, study design, and the steps taken to condition and process the raw data before analysis. Data sharing, on the other hand, emphasizes accessibility and ease of use for secondary analysis. A dataset intended for sharing is thus necessarily lightweight and is ideally structured for easy use, rather than completeness.

There are several data archival standards in geomorphology-adjacent communities that have seen mixed degrees of adoption. For example, the Sedimentary Experimentalist Network (SEN) provides a data archival standard (Hsu et al., 2013, 2015) that is used by several labs, but there are many research groups generating experimental data that are not archived in this standard. Similarly, the Scientific Variables Ontology (Stoica and Peckham, 2019) and Community Surface Dynamics Modeling System Basic Model Interface (Peckham, 2014; Hutton et al.) offer schema to standardize model interfaces and data that have been adopted by large modeling teams (e.g., Hughes et al., 2022; Patel et al., 2025), but have not yet been integrated into the model development and data sharing workflows of many small research teams. While we cannot know exactly why these frameworks have not been implemented by independent academics, it could be because the specification terminology has many requirements, and users do not clearly see benefit of expending effort to adhere to the specifications.

From conversations with workshop participants, we think that if the goal is to improve data sharing and interoperability, trying to standardize data archival in geomorphology is the wrong approach. Experimental, field, and modeling data in geomorphology are sufficiently heterogeneous that any all-encompassing data standard is too onerous for scientists to adopt into their workflows. Instead, we propose that it would be better to standardize a common derived data product (e.g., Eaton et al., 2024), with focus on ease of implementation and clear expected return on time invested as top priorities.

One of the most common data products that geomorphologists use for analysis is a gridded array populated with values (i.e., a raster). Usually, these arrays have one to three spatial dimensions and a time dimension. We found in our workshop that this basic data type—a time by space by space gridded array—is nearly universally used by geomorphologists that work across modern observations, experiments, and numerical models. Everyday examples include satellite images (2D spatial array with a time dimension that has one value) or a time-series of topography from a flume experiment (2D spatial array with a time

dimension having many values). Based on our workshop, consensus emerged that a data object of this type should form the core of an analysis-ready data standard in geomorphology.

## 2.2 A geomorphology and sedimentology analysis software tool must be modular.

The heterogeneous nature of gridded information used in geomorphic analysis—for example, digital elevation models, laboratory overhead imagery, satellite imagery, and numerical simulation elevation models—presents a challenge to creating a single unifying analysis workflow. A clear answer to this challenge that emerged in the workshop was modular software. Akin to building with interchangeable blocks, modular software provides the infrastructure and tools to complete analysis but allows the user to choose how the tools fit together. For example, a researcher might need to invent a pipeline to process a combination of field data sources into a binary mask of a river, but once that mask is derived, the researcher could leverage a modular software solution implementing common analytical routines for feature extraction, spatial analysis, and statistical characterization of channel morphology and networks.

There are many overlapping benefits to modular research software. A modular design would encourage algorithm reuse, and less obviously would enforce reproducible science by explicitly using the exact same code, not just the same idea (Grieve et al., 2020). In this way, standard steps in an analysis workflow can leverage built-in software functionality, but innovations requiring custom code can be added transparently, and are therefore easy to identify and review separately from standard analytical steps. Modular software also breaks analysis into pieces that are more easily scaled for parallelization and distributed computing (Dask Development Team, 2016), which become important considerations when working with high-resolution data and multiple realizations of an experiment or model to assess uncertainty. Lastly, modularity enables interoperability with established geomorphological tools and models (e.g., Schwenk et al., 2017; Adams et al., 2017; Müller et al., 2022) and widely used GIS software, which workshop participants emphasized was an important feature they would expect to have in a software package for geomorphic analysis.

## 2.3 A lower-cost experimental hardware system is needed to sustain this mode of inquiry.

State-of-the-art experimental geomorphology facilities are concentrated in a few laboratories around the world, due to high space requirements and high construction and operation costs. Additionally, many existing systems rely on deprecated and vulnerable firmware/software and use outdated hardware that is no longer supported or manufactured, making these systems exceptionally fragile in the online and connected world. Concentrated facilities and high barriers to entry mean that innovative hypotheses go untested if researchers cannot easily gain access to a suitable lab facility. There are two potential solutions to this problem. One model used by other disciplines like analytical chemistry is to develop expensive lab resources as community shared facilities where visiting scientists can come and use existing equipment (e.g., flumes or tanks or scanners). However, it is not as easy to book time in a flume as it is to bring samples to advanced analytical facilities (microscopes, microprobes, and spectrometers); geomorphological experiments almost always require building a custom apparatus and novel workflow. Another solution is to develop low-cost and modular systems so that researchers can use small budgets to build a less advanced facility for themselves. The key required components of these experimental systems that drive up the cost are (1) imaging and

sensors, (2) motion control for automating positioning, (3) plumbing and sediment flow control. Off-the-shelf motors, sensors, cameras, and control systems have become widely available and standardized in the last decade due to advances in 3D printing, CNC manufacturing, and logistics automation. These advances have not been widely integrated into geomorphological and sedimentological experimental setups. Experimental systems could be made more affordable, adaptable, and maintainable by leveraging off-the-shelf materials (e.g., aluminum extrusion, stepper motors), and existing community-supported open-source frameworks (e.g., Arduino; Barragán, 2004; Shiloh and Banzi, 2022).

Low-cost experimental facilities could substantially democratize access to geomorphological experimentation and enable science currently out of reach for many labs. Though there is a tradeoff between cost and accuracy, off-the-shelf industrial control components are more capable and easier to use than ever (Wenzel, 2023; Nishio et al., 2025). We expect that most geomorphology applications will be well-served by currently available low-cost technology. In any case, hardware systems should be built for modularity and extensibility; it is difficult to anticipate future needs and technologies, and experimental apparatuses should be able to evolve without needing a complete rebuild. Finally, access to community-designed experimental hardware will also distribute system maintenance responsibility across the community, freeing up engineer time for new innovation.

## 2.4 Building a community around open-source tools in geomorphology depends on a system to pass credit along to creators.

Recognition of intellectual contributions remains a primary evaluation criterion for candidates in academic job searches and promotion processes (e.g., Schimanski and Alperin, 2018). There is a crucial need for any community-maintained system of open-source tools in geomorphology to incorporate mechanisms that explicitly recognize and credit user contributions (Buttliere, 2014; Smith et al., 2016; Stirling, 2024; Katz et al., 2021). This point was also notably raised during the Sediment Experimentalists Network efforts (Hsu et al., 2013, 2015), which acknowledged the importance of properly attributing contributions within collaborative scientific endeavors.

The Community Surface Dynamics Modeling System (CSDMS; Tucker et al., 2022) and Landlab modeling toolkit (Hobley et al., 2017; Barnhart et al., 2020) include automated tools to generate a citation list based on data and code components integrated into a user's workflow. Including references in algorithm documentation will highlight intellectual sources at the time of use, and a human-readable citation file at the repository level can point users to appropriate software references. A transitive credit system may enable credit to be automatically passed to software creators (Katz and Smith, 2015), who typically do not receive credit for their contributions through traditional citation mechanisms. Any framework for data sharing and analysis software in geomorphology should build on these best practices, and create frictionless mechanisms to acknowledge algorithm and data sources.

## 3 First links in the *sandpiper* toolchain

In response to these community needs, and guided by the feedback from workshop participants, we have launched a cyberinfrastructure toolchain—called *sandpiper*—with the objective to improve code and data reuse in geomorphology (https:

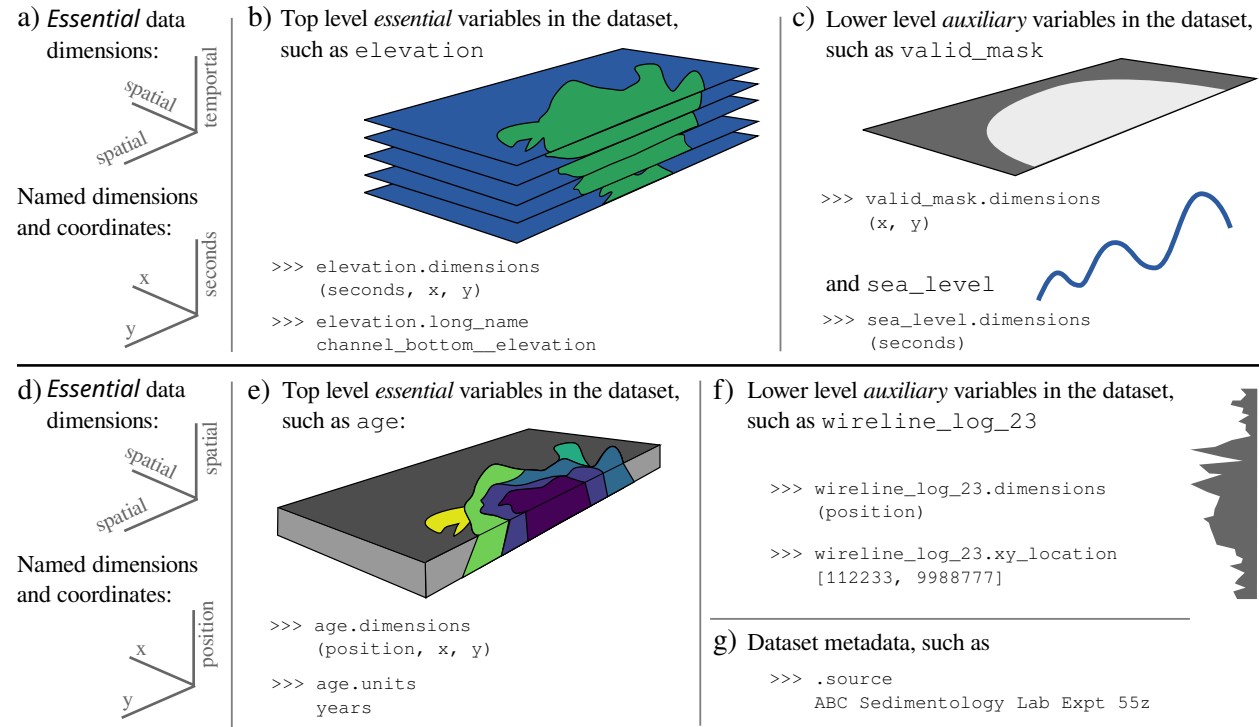

**Figure 1.** Example three dimensional time-by-space-by-space experimental delta dataset (a–c) and space-by-space-by-space experimental delta dataset (d–g), both implemented to follow the *sandsuet* data specification. a) Essential data dimensions for a spatiotemporal dataset, and the corresponding coordinates. The dataset is organized with the temporal dimension first. b) Example of essential data, which has spatiotemporal dimensions and a brief metadata description (Stoica and Peckham, 2019). c) Example of auxiliary data, which has fewer than three dimensions, and supports analyses using the essential data. d) Essential data dimensions for a three-dimensional spatial dataset (i.e., stratigraphy), and the corresponding coordinates. e) Example of essential data, which has three spatial dimensions. f) Example of auxiliary data, which has fewer than three dimensions. g) Example of metadata existing at the dataset level.

//github.com/sandpiper-toolchain). This broad effort is multi-pronged. Major components of the project are to (1) establish a data storage structure and specification that facilitates data sharing in geomorphology, (2) create modular software promoting algorithm reuse, (3) build open-source hardware kits for do-it-yourself sedimentary experiments, and (4) establish a credit mechanism such that developing community tools engenders recognition and career advancement. This toolchain is designed to be easy to adopt, flexible and scalable to engage a wide swath of researchers, and powerful enough for collaboration and data/tool sharing. With *sandpiper*, we aim to transform geomorphologists from solo hackers into a community of users who support software product development as a part of their normal research workflow.

## 3.1 *sandsuet*, an analysis-ready data specification

We have created a data specification for analysis-ready and shareable geomorphology data. This specification is not an archival standard; there exist robust schemata for archiving geomorphology data (e.g., Hsu et al., 2015). Instead, this specification gives a minimum set of criteria that rasterized geomorphology data need to be immediately usable for analysis. The specification follows a community-built consensus on the kinds of data we want to be sharing, and how we want to be sharing that data.

We call this data specification *sandsuet*. *sandsuet* v1.0.0 is designed specifically for rectilinear gridded data in projected coordinate systems. The *sandsuet* specification is agnostic to the file storage format, but is inspired heavily by the NetCDF4 schema (Rew et al., 2006). In fact, *sandsuet*-compliant data can be thought of as a subset of NetCDF4-compliant data, with a few additional restrictions to make working with geomorphology data simpler. NetCDF4 is the recommended format for *sandsuet*-compliant data, but it is not strictly required. For example, data could be grouped in hierarchically-organized computer directories to make a compliant dataset, but this approach would be more cumbersome than using NetCDF directly.

The *sandsuet* specification applies to data up to four dimensions (up to one temporal dimension, and up to three spatial dimensions), and allows any number of variables that record arbitrary scalar values to populate the dimensions as gridded arrays. All variables must be located in space and/or time by coordinates, and must be designated with units and a brief description. However, *sandsuet* does not attempt to standardize variable or dimension names (Eaton et al., 2024). The required metadata enables variables to be easily understood, while also allowing for users to specify variable names and dimensions according to their preference and without undue burden.

The *sandsuet* specification distinguishes between essential data meant to be shared and auxiliary data that support the essential data (Figure 1). Building on the hierarchical organization of NetCDF4 (Rew et al., 2006), *sandsuet* specifies that auxiliary data be placed into one or more "groups" that are organized hierarchically lower than the essential data. Auxiliary data includes all variables with lower dimensionality than the essential data, like a timeseries of sea level in a spatiotemporal dataset of an experimental delta, and can also include data with equal dimensionality to essential data, like masks of channels in a delta. Beyond classification based on dimensionality less than essential data, it is up to the dataset creator to determine which data are essential and which auxiliary; examples of auxiliary variables are given in Figure 1c, f.

Both essential data and auxiliary data variables can (and should) include metadata that provide context to the data (Hsu et al., 2015). Metadata are conceptually distinct from auxiliary data, insofar that metadata describe data, rather than support analysis of data. Examples of variable metadata are data units (required), data collection instrument model name/number, and some score of measurement bulk error. The dataset can also have metadata, such as a description of where and when data were generated/collected, who compiled the data into the *sandsuet* specification, and the *sandsuet* version number (required).

We suggest that *sandsuet* be implemented into existing data management and sharing workflows at the data processing and analysis stage (Phase 5 of the experimental geomorphology lifecycle; Hsu et al., 2015). Data cast in the *sandsuet* specification have therefore been quality checked and, if applicable, minimally processed for further analysis (e.g., small missing-data areas filled by interpolation); the appropriate amount of data cleaning and processing is determined by the dataset creator. A benefit of beginning analyses from data already prepared for sharing is that study conclusions are fully reproducible (Hsu et al., 2015).

When data creators are ready to share their *sandsuet* data, placing the file in a public and persistent repository (e.g., Zenodo) with tags to make the data findable is considered a best practice.

Data standardization requires deliberate decisions about what are common data attributes, how to define those attributes, and which should be mandated or permitted; these decisions both define and limit the scope and flexibility of the standard. *sandsuet* does not require datasets to include measures of uncertainty or error; we argue that most geomorphology data sharing use cases benefit from compactness and simplicity, and that flexibility in the specification allows a dataset creator to include auxiliary variables that convey uncertainty if needed. The *sandsuet* specification requires gridded data, and is therefore not compatible with geographic coordinate systems and/or unstructured meshes. Recognizing that community needs for an analysis-ready data specification may change in the future, *sandsuet* uses Semantic Versioning (https://semver.org/) to describe backwards-compatibility of files formatted according to the specification.

The complete *sandsuet* specification is given in the Appendix, including links to data examples that are compliant with the specification. The linchpin of any solution to reproducible and easy data sharing in geomorphology is a shareable data specification, and we expect this data sharing specification has the potential to significantly lower barriers to replicating research results and accelerating science in geomorphology.

### 3.2 Community consensus should guide development addressing community needs

The *sandsuet* data specification offers a solution to data sharing challenges in geomorphology, and addresses one of four key insights identified during a community workshop. Progress is still needed towards solutions addressing open-source hardware and software needs in geomorphology, and mechanisms for distributing credit to open-source developers.

The *sandpiper* toolchain can serve as the hub for developments addressing remaining community needs (Katz et al., 2018). Importantly, toolchain components should keep the community at center, and depend on community buy-in to be successful (Boettiger et al., 2015; Katz et al., 2018). To encourage community participation and guide developments, the *sandpiper* toolchain has established a governance structure and Code of Conduct (https://github.com/sandpiper-toolchain/governance) that are able to grow with evolving community suggestions and needs (Boettiger et al., 2015). Toolchain developments in community spaces (e.g., https://github.com/sandpiper-toolchain) will enable credit to be effectively passed on to creators.

### 4 Conclusions

Here we report on the findings from a community-needs workshop where we developed a framework for improving data sharing, increasing software reuse, and democratizing experimental approaches in geomorphology. Workshop participants identified the need for a data organization schema to facilitate data sharing. In response to the findings in that workshop, we report a newly developed specification for a shareable data schema based on NetCDF4 that directly serves a common need in geomorphology research. This schema, called *sandsuet*, is designed to be flexible enough to accommodate most kinds of rasterized geomorphology data. However, the *sandsuet* specification prescribes some dimensional and labeling requirements

that structure and organize rasterized data in a way that intuitively simplifies input into geological analysis workflows. *sandsuet* provides a way to organize new data for publication and a way to give new life to old data sets.

*Code and data availability.* No new data were generated for this publication. Appendix A reproduces the *sandsuet* v1.0.0 text verbatim, with the version of record text archived on Zenodo at 10.5281/zenodo.17884433. Example data and example codes for generating new *sandsuet* datasets are available at https://github.com/sandpiper-toolchain/sandsuet.

## Appendix A:  Analysis-ready Geomorphology Data Specification — *sandsuet* v1.0.0

1. The dataset shall be organized hierarchically.

    (a) The top level shall contain variables that (i) have the maximum number of dimensions in the dataset and (ii) make up the essential data that the creator intends to share for reuse.

    (b) Lower levels, if present, shall contain data and information that reference and support the essential data variables, hereafter referred to as auxiliary data.[1]

    (c) Lower level group names are arbitrary.

2. All data shall be arranged in a rectilinear grid and projected.

3. The dataset shall have attributes briefly describing and contextualizing the underlying data; the only required metadata field is the "sandsuet_version". [2]

4. Data dimensions shall be ordered according to the hierarchy: temporal > vertical spatial > horizontal spatial > horizontal spatial. If any dimensions are not present in the underlying data, that dimension is to be simply omitted.

    (a) Ordering of horizontal spatial dimensions is arbitrary. However, if spatial information represents real-world information, the north-south oriented dimension shall be ordered first (e.g., UTM Northing).

    (b) No more than four dimensions are permitted for a single variable.

    (c) Spatial dimensions shall be orthogonal to one another.

    (d) Dimension names are arbitrary. [3]

5. A coordinate variable shall be specified for each dimension represented in the dataset.

    (a) Coordinate names shall match the corresponding dimension name exactly.[4]

    (b) Coordinates shall provide specific rectifiable information to place the data along applicable dimensions.

    (c) Spatial coordinates must have uniform spacing, and temporal coordinates may have non-uniform spacing.

    (d) Coordinates must be monotonically increasing or decreasing, with the exception of temporal coordinates, which must be monotonically increasing.[5]

6. Variables shall be labeled with applicable coordinates.

    (a) Any number of coordinates can locate variables along the same data dimension.[6]

    (b) Variable units shall be consistent across the dataset.[7]

    (c) Variable attributes shall specify variable units and a description of the variable information.[8]

    (d) Variables shall have between zero and four dimensions (scalar up to time by three spatial dimensions).

    (e) Variable names are arbitrary.[9]

    (f) Missing values shall be filled consistently across a variable, and variable metadata shall indicate the fill value.[10]

---

[1]Note, auxiliary data is distinct from metadata; see Specification 3 and 6c.

[2]I.e., metadata. Metadata are information *about* the dataset as a whole and/or individual variables. For example, date generated, author, DOI of an associated publication, variable units, or which instrument generated the variable data. The NetCDF attribute "sandsuet_version" shall be specified without preceding "v", e.g., 1.0.0.

[3]Consider that descriptive dimension names are helpful. For example, a temporal dimension could be named "elapsed seconds", "Date", or "Myr".

[4]This forms a dimension-coordinate pair that is essential for referencing data in absolute space and time, that is, in relation to other variables in the dataset (e.g., Specification 6a).

[5]For example, use elevation, rather than depth or two-way travel time for stratigraphic information.

[6]For example, if Sensor 1 data collected every 5 minutes, and Sensor 2 data collected every 20 minutes, each should have their own temporal dimension name and matching coordinate variable, and each should reference the same absolute time (e.g., elapsed time since beginning of experiment; Specification 6b).

[7]For example, if one spatial dimension has units of meters, additional spatial dimensions should also use meters, including when multiple dimension-coordinate pairs locate data along the same dimension. Derived variables, like velocity, should also then use meters, as meters per second. Consider applying the UDUNITS specification.

[8]Consider the Scientific Variables Ontology (Stoica and Peckham, 2019) for the NetCDF "long_name" variables attribute.

[9]Consider using memorable and easy to type variable names that are aligned with conventions of the discipline.

[10]For example, the NetCDF specification indicates missing values with a variable attribute called "_FillValue". Different variables can use different fill values, if needed.

*Author contributions.* AJM: conceptualization, methodology, investigation, writing (original draft preparation), writing (review and editing). EB: conceptualization, methodology, investigation, writing (original draft preparation), writing (review and editing). EH: conceptualization, investigation, writing (original draft preparation), writing (review and editing) CN: conceptualization, writing (original draft preparation), writing (review and editing) ADW: conceptualization, writing (original draft preparation), writing (review and editing) JM: conceptualization, writing (review and editing)

*Competing interests.* The authors have no competing interests to declare.

*Acknowledgements.* We thank the National Science Foundation Office of Advanced Cyberinfrastructure (OAC) for supporting the authors, the workshop that generated reported community needs, and the sandpiper framework (Awards #2411033, #2411034, #2411035, and #2411036). Jayaram Hariharan helped develop an early standardization of three-dimensional datasets that eventually led the authors to pursue development of a sharable specification. We are grateful to the participants of the April 2025 sandpiper workshop and the May 2025 Community Surface Dynamics Modeling System meeting clinic, for their role in creating and testing the *sandsuet* data specification, and generating a shared vision for open-source software and hardware for the community. We also thank Stuart Grieve and an anonymous reviewer for their thoughtful feedback on an earlier draft of this manuscript.

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
