# Peer review of "Short Communication: The need for open-source hardware, software, and data-sharing specifications in geomorphology"

_EGUsphere, 2025_

## Author Comment (AC1)

**Author responses in blue**

https://egusphere.copernicus.org/preprints/2025/egusphere-2025-4770/

**Reviewer 1 - Stuart Grieve**

I'd like to thank the authors for this manuscript. Work to improve reproducibility, openness and data sharing within geomorphology research feels increasingly important in a time when data volumes are increasing and funding in many countries is contracting. This manuscript describes the outcomes of a recent workshop where geomorphologists came together to discuss key challenges within the geomorphology community, which were identified as: data archival and sharing; modular software; open hardware and academic credit. The authors propose a new framework, sandpiper, to begin to address these challenges and the remainder of the manuscript describes a data specification, sandsuet, which addresses the first of these concerns. The manuscript is clear and well written and sets a good balance between articulating the challenges we as a community face, while moving us towards potential solutions.

Thank you for a positive and constructive review. We are so glad to hear feedback affirming the overall approach, and appreciate the suggestions to both improve the manuscript and guide our future work.

General comments

Given the nature of this manuscript I do not have significant requests for additional work or analysis, but rather have a some questions and observations that I hope can help the authors to strengthen the work.

1) The first line of the abstract talks about geomorphologists having more data and compute than ever before. Do the authors believe that this is a problem that is unique to geomorphology, or that given the nature of our work, presents in a unique manner relative to other disciplines? I think there is scope to expand the introduction by giving some context from other fields outside the geosciences, identifying solutions that may exist and be appropriate for us to adopt, and highlighting where our disciplinary context makes that difficult. We wrote about data, code and reproducibility a few years back (Grieve et al., 2020) and you might find some useful references within that work to help frame these ideas.

We appreciate this comment, since it's something we talk about often. The climate modeling community, in the effort to make CMIP and other inter-model comparisons, had to tackle this problem a long time ago. In fact, we take a lot of inspiration from those efforts. We have added extra text in the beginning to sketch out some of that context.

2) I am really pleased that you are building this solution on top of NetCDF rather than developing a new standard from scratch. But when reading the manuscript I got really far through before this became apparent. Some foregrounding of NetCDF, and its wide adoption and use in other disciplines would help get people up to speed more quickly.

This is a good point. In the geomorphology world, NetCDF is not widely used, so we originally chose not to introduce it early in the paper and frame the problem first. Considering your comments, we agree this buries the lede, and have introduced NetCDF earlier in the text.

3) The manuscript talks throughout about building community, which I agree is vital for any effort such as this. There has been a lot of work done in the Research Software Engineering community around how to build and sustain communities around software projects. One nice example comes from the R community (Boettiger et al., 2015) and there is the work of Katz et al. (2018) taking a broader view of things. There are also a lot of more general resources on the Software Sustainability Institute's site: https://www.software.ac.uk/resource-hub These might help the authors frame this need within the context of what can be achieved, and what resources are needed to achieve it.

Thanks for the link to resources! We will certainly incorporate these ideas as we experiment with our community-building efforts. We have not modified the text to reflect this, but it will inform our ongoing work as we build out these tools.

4) Similarly, in section 2.4 the authors discuss scientific credit, which is indeed very important in ensuring that everyone that contributes to a project is being recognised. There is a big body of research on software citation, both understanding how and why people cite or don't cite software, but also looking at practical solutions that make it easier to cite software or other "non-traditional" outputs. One example of this would be the citation file format, and it's integration into platforms like github: https://citation-file-format.github.io/ as well as the broader work of the FORCE11 Software Citation Working Group (Smith et al., 2016; https://force11.org/group/software-citation-working-group/). Some other relevant recent work on this topic includes Katz et al. (2018, 2021).

We appreciate this suggestion, and we've decided to try and apply it to this product as well, by placing a .cff citation file in the *sandsuet* repository that points to the archival Zenodo repository (https://doi.org/10.5281/zenodo.17884433) and will point to this ESurf article pending acceptance and a permanent doi becoming available. In this manner, *sandsuet* can be referenced by DOI as both a software product and a peer-reviewed article.

We have added several references to the text, based on your suggestions.

Line 43: "and needs in a standards" I struggled to parse this sentence.

Thank you for pointing out this difficult to read sentence. We have split the sentence in two, to make it more readable.

References

Boettiger, C., Chamberlain, S., Hart, E., & Ram, K. (2015). Building software, building community: lessons from the rOpenSci project. Journal of open research software, 3(1), e8-e8.

Katz, D.S., McInnes, L.C., Bernholdt, D.E., Mayes, A.C., Hong, N.P.C., Duckles, J., Gesing, S., Heroux, M.A., Hettrick, S., Jimenez, R.C. and Pierce, M., 2018. Community organizations:

Changing the culture in which research software is developed and sustained. Computing in Science & Engineering, 21(2), pp.8-24.

Katz, D.S., Hong, N.P.C., Clark, T., Muench, A., Stall, S., Bouquin, D., Cannon, M., Edmunds, S., Faez, T., Feeney, P. and Fenner, M., 2021. Recognizing the value of software: a software citation guide. F1000Research, 9, p.1257.

Katz, D.S. and Chue Hong, N.P., 2018, July. Software citation in theory and practice. In International Congress on Mathematical Software (pp. 289-296). Cham: Springer International Publishing.

A. M. Smith, D. S. Katz, K. E. Niemeyer, and FORCE11 Software Citation Working Group, "Software citation principles," PeerJ Comput. Sci., vol. 2, no. e86, 2016 [Online]. Available: https://doi.org/10.7717/peerj-cs.86

-- Stuart Grieve

**Reviewer 2**

This paper describes sandsuet, a standard for saving geomorphic raster data sets. The authors distinguish between data sharing and data archiving. My understanding of the difference is that data for sharing has the minimum possible information and is for quick dissemination of data. Data archiving is for storing important data sets which one may want to preserve for a long time so more information would be provided than would be provided with shared data. I don't really know when one gets to the archival stage - maybe after publication? To be honest I had never recognized the two data needs as separate, but after reading it makes sense why more a more flexible data sharing standard is needed.

I think this paper represents the best of our community. I am grateful to the authors and workshop participants for developing sandsuet and presenting it to the community. These types of contributions are generally thankless, as the authors indirectly point out by stating that we need better ways to give credit to open source developers. Ultimately, working on things like this means less time working on the science that is typically recognized when it comes to raise time. But this is an important contribution!

If this data sharing standard catches on it will mean that 1) I can use sandsuet without thinking about how I might want to save my data in the short term. 2) If I write some code to analyze my data saved in sandsuet format, then I can easily apply my code to analyze other people's data that are saved in sandsuet format. 3) If I share my data analysis tools/code, that is a double win for the community. If sandsuet catches on, there would likely be less recreation of analysis code, assuming I/others share my analysis code in a way that is actually useable to others. But that is a different topic not addressed here.

So this is great and please publish it. I have some minor comments. Hopefully they will help the authors see the stumbles that a newbie might have when attempting to use sandsuet. Also, what does sandsuet mean? I wondered why they chose this name.

Thank you for the constructive and positive comments! We are glad you see the value in the standard, and we hope it accelerates your work. As for the name, we will likely not include the story in the paper. But for your interest, we chose the sandpiper toolchain name to reflect a mission to dig into data that often amounts to a pile of sediment (sand), much like a sandpiper family of bird species do to find food. We named the specification after a commonly cube-shaped (like our data) food considered a healthy high-energy treat for birds called suet, because we imagine working with data following the specification will be a delight! To make the specification more uniquely identifiable and easily associated with the sandpiper toolchain, we combined these ideas to give "*sandsuet*".

Everything about sharing of raster data, what absolutely needs to be provided, and the definition of auxiliary variables all made sense to me. What confused me a bit is the statement that sandset is agnostic to the file storage format (L 151). I have never used netcdf, but I know a tiny bit of coding. I found Figure 1 very helpful and imagined that elevation was an instance of the sandsuet class. Later I went and looked at the demo to try and better understand, and I realized that (I think) elevation would be of type netcdf.dataset. I guess I could create my own data class

if I really wanted to. I guess if I didn't use Python I could use Fortran or whatever language. But the whole time I was reading I was hung up on the file storage format. Would this really work if someone saved their data in excel? What if someone saved their data in text files (e.g. esrii ascii format). Maybe it is beyond the scope of this paper, but I thought it would be helpful to address how this might work when different scientists share data in different file storage formats.

We really appreciate this question, because it gets to a very important but subtle point. The standard is inspired by NetCDF. Ultimately, the software we write will use tools built on top of NetCDF to internally handle data. That said, the input data does not necessarily have to use the NetCDF format. You could technically make a *sandsuet*-compliant dataset with a set of folders and text files. It would be a bit more cumbersome to work with initially, but if it obeys the *sandsuet* specification, it will have all the parts in the right arrangement, so that someone will at least have a clear road map as to how to import the data into their workflow. We have added a bit of text to make this subtle distinction a bit more clear.

I also wondered how one researcher might access the sandsuet data of another researcher. I don't think this was addressed. Do I need to know that Jane Doe has data that I want? Will Jane Doe post this on a list serve? Do I need to email Jane Doe? I know this is a tricky issue. I just thought it might be addressed.

We explicitly chose not to specify any place where *sandsuet* datasets need to be stored to be considered "*sandsuet* compliant". Although not in the spirit of the *sandsuet* specification, a researcher can put their data in *sandsuet* format, and retain the data privately without sharing. So, *sandsuet* datasets can be put anywhere, but a best practice would be to place data somewhere public and persistent. The authors are regular users of FigShare, Zenodo, and institutional repositories, and many journals are now requiring data hosting to meet certain requirements. Ultimately, it will be every researcher's responsibility to make sure their data is findable. We have added a sentence to specify this as a best practice.

The authors and the *sandpiper toolchain* do plan to provide tools that make it easy to search for data on the web that data owners tag as "*sandsuet*-compliant." Hopefully if you read a paper that has data in it that you think is interesting, our vision is that in the future, you could follow a link to the correct repository and download the dataset directly, without having to email Jane Doe. We consider establishing this system to be outside the scope of this article, which introduces several needs in the community and establishes the data specification.

I guess this paper must be the first paper coming out of the described conference. In that way, it served a dual purpose - to describe the point of the conference and to present their solution to address one aspect of the conference, data sharing. Personally, I found the discussion about open source hardware, software, and credit to open source developers slightly out-of-place because really this paper is about data sharing. I was excited to learn about shared experimental facilities, but that is not what this paper is about. I would have preferred more discussion of file storage formats and how these data would be shared, rather than a tease about community-designed experimental hardware and shared facilities. I also understand that authors may need to use this paper to meet multiple goals.

It is true that this manuscript meets multiple goals. We certainly agree that a bit more space can be devoted to spelling out some of the details vis-a-vis file formats, etc.; we have added some text to this effect in the Introduction and *sandsuet* specification Sections. The reviewer is right that this paper is the beginning of a series that will continue to come out of this workshop and others that we are planning, and so we think this text remains important to include to make sure we communicate the findings from the workshop participants.

Some tiny comments.

Figure 1e : I think, but I'm not sure, that if grid location x,y is purple at the top, it does not need to be purple at depth. Because the exposed depth side shows no variation with depth, it might incorrectly imply that age cannot vary in z. If it can vary with z, then maybe show that in the example map?

Thank you for noticing this simplification in our sketch. You are correct that age can vary over depth and we have adjusted the figure to better reflect this possibility. This will help readers better understand what the data represent and how their data may fit into the specification.

Line 154 - Does time have multiple dimensions? Is this something everyone knows about but me?

We mean this mostly to say that the time dimension is, strictly speaking, optional.That is, "up to one" temporal dimension allows for the possibility of no temporal dimension.

Line 221 - I guess the designers are trying to be as flexible as possible, but why not impose the ordering of the spatial dimensions? It would be easier if everyone ordered in the same way, and it's not a huge ask. Maybe I don't understand what that means.

We do impose an ordering, but only if the information is oriented in the real world. Many results from numerical simulations or from experiments do not have a notion of north and east, so we leave it up to the user to choose which is which, since the same data could be rotated and still make just as much sense.

Line 225 - dimension names are arbitrary was confusing to me. They aren't exactly arbitrary. They should be descriptive enough that others understand what they represent, right?

We have found through experience (and conversations at our workshop affirmed this), that standardizing names is surprisingly difficult. For example, the Climate and Forecast Data conventions (Eaton et al., 2024) does not standardize dimension or variable names. So while names are arbitrary, it is of course helpful to use intuitive and descriptive names, as you have suggested.

We added a sentence to the relevant footnote suggesting to "Consider that descriptive dimension names are helpful." And, we added a sentence to the main text mirroring the sentence from the CF-convention that "*sandsuet* does not attempt to standardize variable or dimension names".

**References**

Eaton, B., Gregory, J., Drach, B., Taylor, K., Hankin, S., Caron, J., Signell, R., Bentley, P., Rappa, G., Höck, H., Pamment, A., Juckes, M., Raspaud, M., Blower, J., Horne, R., Whiteaker, T., Blodgett, D., Zender, C., Lee, D., Hassell, D., Snow, A. D., Kölling, T., Allured, D., Jelenak, A., Soerensen, A. M., Gaultier, L., Herlédan, S., Manzano, F., Bärring, L., Barker, C., and Bartholomew, S. L.: NetCDF Climate and Forecast (CF) Metadata Conventions, https://doi.org/10.5281/zenodo.14275599, 2024